# Cinacalcet Efficacy in Hyperparathyroidism—Chronic Kidney Disease—Non-Dialysis, Hemodialysis, Peritoneal Dialysis, Kidney Transplantation: Critical Review

**DOI:** 10.3390/biomedicines14010016

**Published:** 2025-12-21

**Authors:** Dominik Lewandowski, Miłosz Miedziaszczyk, Katarzyna Lacka, Ilona Idasiak-Piechocka

**Affiliations:** 1Students Research Group of Transplantation and Kidney Diseases, Poznan University of Medical Sciences, 60-352 Poznan, Poland; 2Department of Clinical Pharmacy and Biopharmacy, Poznan University of Medical Sciences, 60-806 Poznan, Poland; 3Department of Endocrinology, Metabolism and Internal Medicine, Poznan University of Medical Science, 60-355 Poznan, Poland; 4Department of General and Transplant Surgery, Poznan University of Medical Sciences, 60-806 Poznan, Poland

**Keywords:** cinacalcet, chronic kidney disease, hyperparathyroidism, kidney transplantation, hemodialysis, peritoneal dialysis

## Abstract

Hyperparathyroidism is a serious complication of chronic kidney disease (CKD) and can occur in patients not on renal replacement therapy, during dialysis therapy, or after kidney transplantation. The disease leads to an increased risk of cardiovascular events, bone loss, and fractures. Cinacalcet is a widely used drug, but its effectiveness in treating hyperparathyroidism in selected stages of chronic kidney disease remains unclear. This critical review aims to integrate findings from meta-analyses and clinical trials to assess optimal therapeutic strategies in patients suffering from CKD, who are non-dialysis-dependent, dialysis-dependent, and after kidney transplantation. The authors reviewed eligible studies, including meta-analyses, randomized controlled trials, and observational studies assessing biochemical outcomes, cardiovascular, bone, and survival outcomes with cinacalcet. Cinacalcet effectively reduced serum parathyroid hormone (PTH), calcium, and phosphorus across all CKD stages, particularly in hemodialysis patients. Combination therapy with vitamin D analogs enhanced biochemical control without increasing adverse events, although mild, transient hypocalcemia and gastrointestinal symptoms were common. In kidney transplant recipients, parathyroidectomy achieved greater normalization of PTH and calcium. Cinacalcet has been shown to reduce mortality in patients on hemodialysis and peritoneal dialysis.

## 1. Introduction

Chronic kidney disease (CKD) is a global problem affecting approximately 10% of the worldwide population. Patients suffering from this disease experience significantly increased mortality—mainly because of cardiovascular events, which are strictly associated with alterations in calcium–phosphate metabolism [1]. Impaired renal function leads to decreased phosphate excretion, which leads to elevated levels of fibroblast growth factor 23 (FGF23), impaired 1α-hydroxylase activity, and impaired calcium absorption. This leads to excessive stimulation of the parathyroid glands, resulting in increased parathyroid hormone (PTH) secretion. Clinically secondary hyperparathyroidism (SHPT) is characterized by hypocalcemia, hyperphosphatemia, and increased PTH serum level; however, it is also related to vascular calcification, osteoporosis, heightened fracture risk, and immune dysfunctions [2]. In some cases, long-term overstimulation of parathyroid glands causes growth and development of nodules, able to autonomously secrete PTH due to specific pathology, and are much less vulnerable to pharmacological treatment [3].

There are several groups of drugs used in the treatment of SHPT, and vitamin D analogs are effective in lowering PTH levels in CKD patients on dialysis. According to the 2017 KDIGO Guidelines, they are, however, not routinely recommended in pre-dialysis patients, due to high occurrence of hypercalcemia (up to 40%) [4]. One of the reasons explaining such a high frequency of this adverse event is vitamin D toxicity, resulting from the increased intestinal absorption of calcium and mobilization of calcium from bones. Even the recommended doses of 3200–4000 international units (IU) administered daily for at least 6 months, in healthy subjects, can increase the risk of hypercalcemia in a small proportion of users [5]. Recently, the interest in vitamin D supplementation and self-medication is rising due to the COVID-19 pandemic and claims of improving the outcomes, resulting in uncontrolled use and cases of hypercalcemia in the general population [6]. Another, but much less common, therapeutic option is phosphorus binders. Sevelamer carbonate is approved for the treatment of hyperphosphatemia in patients on dialysis and may be considered in patients with hyperphosphatemia in CKD patients not on dialysis [7]. The introduction of calcimimetic agents changed CKD treatment. These drugs work by activating calcium-sensing receptors (CaSR), leading to reduce in PTH synthesis and secretion, therefore effectively controlling calcium and phosphorus levels [8]. Novel studies suggest that lower PTH levels in dialyzed patients are associated with better hypertension control and lower cardiovascular risk [2]. Cinacalcet is a widely used, oral, calcimimetic drug; however, there are certain safety issues concerning high frequency of gastrointestinal adverse effects [9]. These symptoms can lead to lower adherence and a high rate of treatment discontinuation. Its significant advantage is the fact that it does not cause hypercalcemia, which makes it a potentially safe option for patients, although safety data remains variable across the studies [10]. It is worth mentioning that recent studies prove that combining cinacalcet with vitamin D analogs significantly reduces phosphorus and calcium levels and the calcium × phosphorus product with a smaller occurrence of side effects, which shows a favorable option for SHPT patients [11]. In terms of surgical treatment, parathyroidectomy is generally recommended in patients who have failed pharmacotherapy or have advanced hyperparathyroidism, though comparative evidence remains mixed, and inclusion criteria are crucial for study design. Due to lesser susceptibility to medical treatment, parathyroidectomy is much more often recommended in patients suffering from persistent hyperparathyroidism (PHPT) or tertiary hyperparathyroidism (THPT) [3]. Surgical procedure is associated with greater survival in dialysis patients; however, the introduction of calcimimetic drugs significantly reduced the rate of such operations [12]. The aim of this critical review is to assess cinacalcet efficacy in hyperparathyroidism, because despite existing therapeutic options, there are still no unified ways of treatment.

## 2. Non-Dialysis Patients

The number of studies assessing the efficacy and safety of cinacalcet in non-dialysis-dependent patients suffering from CKD is very limited. There are two randomized, double-blind, placebo-controlled studies and two one-arm observational trials. In this chapter, we would like to summarize their findings.

Charytan et al. performed a single-center study in which 54 patients suffering from stage 3, 4, and 5 CKD were equally randomized into two groups (placebo or cinacalcet). The main reason to conduct this analysis was the observation that, throughout CKD patients not receiving dialysis, SHPT is also a serious problem but is underdiagnosed and inadequately treated. Results of the study showed that most subjects treated with cinacalcet (56%) achieved a significant reduction (30% or more) in intact parathyroid hormone (iPTH) concentrations from baseline. It is noteworthy that the recommended level of iPTH in patients with stage 3 CKD equals 35–70 ng/L, and in patients with stage 4 it is 70–110 ng/L. The study finished with 18 participants in the cinacalcet group and 20 participants in the placebo group. When it comes to safety, the majority of adverse events were not serious, but almost every participant from the cinacalcet group reported nausea or diarrhea. Cinacalcet did not affect estimated glomerular filtration rate (GFR), but the presented study was the first to emphasize the need to diagnose SHPT early and control PTH, calcium, and phosphorus levels in non-dialysis-dependent patients [13]. The second randomized clinical trial (RCT) was performed by Chonchol et al. and included 404 patients from 73 centers. In this study, 74% of participants in the cinacalcet group achieved a significant decrease in iPTH level—defined as over 30%—in comparison to only 28% of participants from the placebo group. Additionally, this analysis assessed a significant decrease in serum calcium level (−8.9%) in comparison to the control group (+0.8%); however, use of cinacalcet increases serum phosphorus level much more (21.4% increase in comparison 6.8% in the placebo group). There were no differences in the results of patients suffering from stage 3 and 4 CKD; the proportion of patients achieving a significant decrease in iPTH was similar regardless of the stage and vitamin D analogs intake. The most common cause of discontinuation of the trial was adverse events—more often in the cinacalcet group; however, their severity in the majority of cases was described as mild [14]. Both studies have consistent conclusions; however, they did not assess the impact of cinacalcet use on vascular calcification and bone parameters.

There were also smaller, one-arm studies performed. Montenegro et al. analyzed prospectively 26 patients with stage 4 and 5 CKD. The authors combined low-dose cinacalcet (almost every patient received 30 mg) with vitamin D analogs and phosphorus binders. The result was a significant decrease in iPTH and serum calcium, with an increase in serum phosphorus; GFR at the end of the study did not vary significantly. Described outcomes suggest considering administering all 3 groups of drugs in pre-dialysis patients with resistant hyperparathyroidism [15]. The last performed study is observational, retrospective, one-arm, and single-center, including 41 patients with stage 3, 4, and 5 CKD. Its strength is long observation time, which equals 36 months and shows long-term outcomes of cinacalcet use. In this study, over 73% of participants had a significant reduction in iPTH levels, and the mean reduction was 50%. Results are generally consistent with previously described analyses: PTH levels decreased by more than 30% in the majority of participants (73.2%), and serum calcium levels also significantly decreased, while serum phosphorus increased from a mean value 3.59 to 3.82 mg/dL. Due to the long-term observation, there is an important fact that at the end of the study: 87.5% of patients were taking vitamin D analogs to avoid hypocalcemia, despite only two of them experiencing this adverse effect [16].

Although cinacalcet is effective in lowering PTH in non-dialysis-dependent CKD patients, clinical evidence regarding mortality or cardiovascular protection is still lacking. There are no studies assessing vascular calcification progression or cardiac events; therefore, biochemical improvement cannot yet be translated into clinical benefit.

In summary, although cinacalcet significantly reduces PTH values in this group of patients, it causes an increase in phosphorus concentration [10], which in this group of patients may lead to an increase in FGF-23 concentration and a decrease in Klotho protein expression [17,18]. Routine use of cinacalcet is not recommended in non-dialysis patients, but it is worth considering in cases of refractory, progressive secondary hyperparathyroidism [14].

## 3. Hemodialysis Patients

Patients suffering from CKD and undergoing hemodialysis are especially vulnerable to alterations in electrolytes and hormonal alterations. This section aims to summarize key evidence on the clinical efficacy and safety of cinacalcet in hemodialysis patients, with particular focus on its potential impact on mortality and major outcomes. Since the broader introduction of calcimimetic agents, scientists have performed multiple studies assessing their efficacy and comparing them to different drugs and procedures. Currently, there are several meta-analyses investigating results of both RCTs and observational studies, but their results are inconsistent; summarizing them in this section can help unify the best way of treatment for hemodialysis patients.

The EVOLVE study was the first clinical trial to compare the effect of cinacalcet with placebo on the risk of death or major cardiovascular event in patients undergoing hemodialysis. The study showed a non-significant reduction (7%) in the risk of the primary endpoint and a reduction in the rate of parathyroidectomies by more than half [19]. Wang et al. were comparing cinacalcet to placebo, based on results from 19 RCTs. An important finding is the fact that cinacalcet was much more effective in lowering serum calcium and phosphorus in patients suffering from stage 5 CKD than in stages 3 and 4. Compared to placebo, it did not reduce all-cause and cardiovascular mortality; however, duration of drug administration was one year. The authors outlined that long-term observational studies are necessary to fully investigate this effect. Taken together, these RCT-based analyses demonstrate that cinacalcet consistently improves biochemical parameters but has not shown a definitive mortality benefit within the limited duration of available trials. This analysis, coherently with the differences analyzed in this chapter, proved that the use of this drug reduced the incidence of parathyroidectomy, indicating that it may be a good choice for patients with contraindications for surgical procedures. Similar to other groups of patients, use of cinacalcet is associated with a relatively high risk of mild AEs like hypocalcemia, nausea, vomiting, and diarrhea. At the end of the study authors raised concerns that relying only on biological markers as primary endpoints may not be suitable to assess the clinical efficacy of the drug; that is the reason we would like to also focus on parameters such as mortality, vascular calcification, or incidence of fractures in our critical review [20]. Meta-analysis performed by Zu et al. answers the presented issues by including not only RCTs, but also high-quality, prospective, observational studies. Results of the study suggest that cinacalcet administration is associated with reduced all-cause mortality (compared to no cinacalcet), which is strictly linked to high levels of serum calcium and phosphorus. Use of cinacalcet had a significant impact on the bone parameters: it not only stopped bone loss, but also increased mineralization, measured in the proximal femur, by 2%. It is, however, worth noticing that it has not stopped bone loss in the lumbar spine. Despite its proven positive effect on bone mineralization, it has no significant effect on reducing the incidence of fractures. These findings suggest that when longer follow-up and real-world data are included, cinacalcet may contribute to improved survival. There are many concerns about the AEs caused by the drug, which are limiting clinicians from prescribing it, but data from this study show that serious AEs are very rare, and cinacalcet-induced hypocalcemia is most often short in duration and self-limiting [21]. Meta-analysis performed by Xu et al. compared the effect of vitamin D alone and vitamin D with cinacalcet; the authors found that combined therapy significantly lowered serum calcium and phosphorus, but not PTH. Authors also focused on AEs; this study proved that cinacalcet with vitamin D did not increase the risk of side effects but had a higher risk of hypocalcemia [11]. In terms of AEs, meta-analysis conducted by Palmer et al. provides valuable insight that out of all calcimimetic agents, cinacalcet has the highest risk of nausea. Authors recommend avoiding this drug in patients wishing to minimize nausea and vomiting risk; however, the median follow-up was 26 weeks—given evidence is only limited to short-term assessment [22]. Lozano-Ortega et al. conducted a meta-analysis whose conclusions are consistent with previous findings; it provided additional evidence that cinacalcet may be associated with a lower risk of all-cause mortality, and emphasized that relying on RCTs is insufficient in detecting differences in clinical outcomes, due to the short period of observation and the long-term impact of the described drug on the cardiovascular system [23]. Certain meta-analyses investigated not only the impact of cinacalcet on serum calcium, phosphorus, and PTH levels, but also different markers. Liu et al. proved that this drug—similar to other calcimimetic agents—reduces osteocalcin levels and increases bone alkaline phosphatase; moreover, they did not find a significant difference in the rate of serious AEs between cinacalcet and placebo [24]. Meta-analysis performed by Liu et al. also showed higher efficacy of cinacalcet than paricalcitol in decreasing the level of calcium–phosphorus product and fibroblast growth factor 23 (FGF23). Importantly, the authors stated that it is difficult to assess which calcimimetic drug has a better ability to decrease PTH, due to individual differences in the sensitivity of CaSR [25].

Contrary to the presented results, there are studies showing a greater impact of parathyroidectomy on laboratory parameters and mortality, compared to medical therapy. Accordingly, surgical treatment is associated with a significant reduction in all-cause and cardiovascular mortality. Additionally, it is more effective in lowering iPTH level: a high level of this marker promotes aortic valve calcification and myocardial injuries. It is, however, an important fact that in many studies, patients submitted to parathyroidectomy were younger and less often diabetic compared to those on pharmacological therapy, showing that monitoring bias of the studies and critically analyzing every patient’s case is crucial to make proper clinical decisions [26,27]. Comparison of effect of cinacalcet in described studies is presented in Table 1. 

Only one described above meta-analysis provided an economic evaluation. It is difficult to assess because of multiple variables between countries, but analysis based on a Japanese observational study showed that cinacalcet contributes to the reduction in costs due to lower rates of parathyroidectomies, cardiovascular events, and hospitalizations, and lower incidence of fractures; however, not every meta-analysis agree with such results. There is still a need for larger assessments considering differences between European and American countries [28]. Palmer et al. also indicated that due to the necessary complex approach to SHPT patients undergoing dialysis, it is necessary to conduct a large meta-analysis to fully assess the effect of parathyroidectomy, calcimimetic agents, phosphorus binders, and vitamin D analogs [22].

Data derived from RCTs failed to show a significant reduction in mortality or cardiovascular events; however, several observational studies report lower all-cause and cardiovascular mortality, suggesting that a benefit is possible during long-term follow-up. Future trials should prioritize clinical endpoints rather than biochemical parameters.

Cinacalcet is recommended for hemodialysis patients. The goal of therapy is to lower PTH levels and reduce phosphate and FGF-23 levels. Caution should be exercised when making therapeutic decisions in the presence of hypocalcemia. In summary, available evidence indicates that cinacalcet effectively controls secondary hyperparathyroidism and may have a favorable effect on survival when used long-term, though definitive evidence for mortality reduction is still lacking. Further large-scale and long-duration studies are needed to confirm its impact on patient survival and cardiovascular outcomes.

In the group of hemodialysis patients, etelcalcetide has been available on the market in Europe since 2016, in addition to cinacalcet. In the available scientific literature, there is one randomized clinical trial comparing the efficacy of cinacalcet (*n* = 343) with etelcalcetide (*n* = 340). Etelcalcetide was more effective in reducing parathyroid hormone (PTH) levels in hemodialysis patients with moderate to severe secondary hyperparathyroidism than cinacalcet. The percentage of patients achieving at least a 30% reduction in PTH levels was 68.2% in the etelcalcetide group, compared with 57.7% in the cinacalcet group. At least a 50% reduction was achieved in 52.4% of patients in the etelcalcetide group and 40.2% in the cinacalcet group. It is important to note that the greater efficacy of etelcalcetide was associated with a higher incidence of hypocalcemia, which was observed in 68.9% of patients treated with the intravenous drug compared with 59.8% of those receiving cinacalcet. Etelcalcetide may be a more effective therapeutic option for the control of secondary hyperparathyroidism in patients undergoing hemodialysis [29]. The drugs differ primarily in their route of administration and pharmacokinetics, which is important for efficacy and use. Etelcalcetide is administered intravenously after hemodialysis and has a long half-life (48–72 h), whereas oral cinacalcet may have variable absorption depending on diet [29]. Etelcalcetide more often causes hypocalcemia, requiring modification of supportive therapy, while the adverse event profile regarding nausea and vomiting is similar for both drugs [29]. Cinacalcet is metabolized by cytochrome P450 enzymes and may interact with other drugs, unlike etelcalcetide, whose metabolism is not dependent on cytochrome P450 enzymes [29]. Intravenous etelcalcetide provides approximately 20% higher adherence compared to cinacalcet, which is important in hemodialysis patients [30]. In cost-effectiveness analyses, etelcalcetide demonstrates an advantage in quality-adjusted life-year (QALY) gains with minor cost differences, and its incremental cost-effectiveness ratio (ICER) is within acceptable limits in European healthcare systems [31]. In turn, real-world studies indicate significantly lower PTH levels in patients treated with etelcalcetide compared to cinacalcet, which is associated with both its pharmacological efficacy and better adherence [31].

## 4. Peritoneal Dialysis Patients

There is limited data in the available scientific literature regarding the efficacy of cinacalcet in secondary hyperparathyroidism in patients on peritoneal dialysis. FGF23 is a protein that regulates phosphate metabolism. Elevated levels of FGF23 are associated with a higher risk of adverse cardiovascular events in patients with chronic kidney disease, including those receiving peritoneal dialysis. Cinacalcet has been shown to significantly reduce serum FGF23 levels [32,33]. Lindberg J et al. performed a randomized, double-blind, placebo-controlled trial to evaluate the efficacy of cinacalcet in patients undergoing peritoneal dialysis (*n* = 34). Cinacalcet was superior to control with respect to PTH reduction outcomes, including the percentage of patients with mean iPTH levels ≤ 300 pg/mL (50 vs. 9%), the percentage of patients with an iPTH reduction of ≤30% from baseline (65 vs. 13%), and the percentage of patients with a reduction of ≤20, ≤40, ≤50% from baseline. Cinacalcet also significantly reduced serum Ca x P levels compared with control (−8.5% vs. 1.4%) [34]. In a prospective observational study (*n* = 33), cinacalcet significantly reduced parathyroid hormone levels after 12 months of study (from 87.5 ± 28.7 pmol/L to 34.5 ± 45.5 pmol/L, *p* < 0.0001) [35]. In another study (*n* = 54), cinacalcet also reduced parathyroid hormone (585 pg/mL vs. 271 pg/mL), Ca (9.80 mg/dL vs. 9.45 mg/dL), and (P 5.80 mg/dL vs. 4.99 mg/dL) levels after 12 months of the study [36]. In a retrospective study (*n* = 27), no significant change in iPTH concentration was observed after a mean of 15.6 months of study (1145 ± 449 pg/mL vs. 1131 ± 642 pg/mL) [37]. A small research group suggests caution in using research results. Phosphorus, FGF-23, and PTH levels correlate positively with both pulse wave velocity and arterial calcification, which are independent predictors of cardiovascular mortality [34]. In two independent studies, no significant differences in PWV were observed after a minimum of one year of follow-up [38,39]. SHPT in chronic kidney disease is associated with cardiovascular events and bone mineral density disorders. Wang A et al. conducted a randomized trial in patients on peritoneal dialysis with secondary hyperparathyroidism (*n* = 65) comparing the effects of cinacalcet and total parathyroidectomy (PTx) with forearm autograft on cardiovascular outcomes (left ventricular mass, coronary artery calcification, heart valve calcification, aortic stiffness, and biochemical parameters and health-related quality of life (HRQOL) indices). Significant reductions in plasma calcium, phosphorus, and parathyroid hormone levels were observed in both study groups. Patients treated with cinacalcet were less likely to be hospitalized for hypercalcemia (1.8%) than those treated with PTx (16.7%) (*p* = 0.005). After 12 months no significant changes were observed in the cardiovascular outcomes studied in either study group [31]. In a subsequent study, the authors compared the effects of both therapies on bone mineral density [40]. Total parathyroidectomy and cinacalcet significantly improved BMD at the lumbar spine and femoral neck at 12 months. The percentage of study participants classified as having osteopenia/osteoporosis based on T-score at the femoral neck decreased from 78.2% to 51.7% in the total parathyroidectomy group (*p* < 0.001) and from 65.7% to 52.0% in the cinacalcet group at 12 months (*p* = 0.7), and at the lumbar spine decreased from 53.1% to 31.0% in the total parathyroidectomy group (*p* = 0.01) and from 59.4% to 53.8% in the cinacalcet group (*p* = 0.3) [41]. In a retrospective study (*n* = 581), cinacalcet treatment was associated with a 54% lower risk of death (HR 0.46, 95% CI 0.31–0.69). Hyperphosphatemia (>6 mg/dL) was associated with an 85% increased risk of mortality (HR 1.85, 95% CI 1.30–2.65) [42]. In patients receiving peritoneal dialysis, cinacalcet is indicated for the treatment of hyperparathyroidism, particularly when PTH and phosphorus levels are elevated. The goal of therapy is to lower PTH and reduce phosphate and FGF-23 levels, which may reduce the risk of skeletal and cardiovascular disorders. Caution should be exercised when making therapeutic decisions in the presence of hypocalcemia [34].

## 5. Kidney Transplant Recipients

One of the therapeutic options to manage PHPT or THPT in patients after kidney transplantation is parathyroidectomy. Despite many advantages of surgical treatment, there are several concerns that are still being discussed: first of all, subtotal parathyroidectomy, being a standard of care in the described illness, is an invasive procedure bearing a risk of complications, and surgical expertise is necessary to correctly assess the amount of parathyroid tissue to remove to avoid hypoparathyroidism [41]. Use of cinacalcet stands as an alternative solution; however, up to this day, there are no clear guidelines on treating PHPT/THPT after kidney transplantation.

We found six studies comparing the efficacy and safety of cinacalcet and parathyroidectomy, but only the study performed by Cruzado et al. is a prospective RCT. It is a multi-center, open-label trial including 30 patients (15 in the cinacalcet and 15 in the parathyroidectomy group). The primary endpoint was normocalcemia at the end of the study, one year after initiation. The authors also focused on secondary endpoints such as iPTH levels, serum phosphate concentration, vascular calcification, bone mineral density (BMD), and renal function. Almost all participants achieved normalization of serum phosphate concentration; however, reduction in iPTH level was greater in the parathyroidectomy group, similar to serum calcium levels; every patient who had undergone surgical treatment achieved normocalcemia compared to 10 out of 15 patients taking cinacalcet. When it comes to renal function, there was a decline in GFR in both groups, but it was greater in patients from the cinacalcet group. There is, however, no proof for an existing correlation between hypercalcemia and higher levels of creatinine [43]. Additionally, in the parathyroidectomy group there was a significant improvement in BMD assessed in the femoral neck observed. Such a change was not observed in patients from the cinacalcet group. Despite all the listed effects, there was no change in the level of vascular calcification in both groups, which suggests that these changes are irreversible and unconnected to the level of laboratory parameters, at least in the assessed period of time. The most common AEs were hypocalcemia in surgical patients and gastrointestinal problems in the cinacalcet group. It is worth mentioning that it was the main limitation to increasing the dose of the drug. Authors also performed a cost-effectiveness analysis stating that in the long term (longer than 14 months), subtotal parathyroidectomy is considered a better option in economic terms [44]. To further analyze the long-term impact of both therapeutic approaches, investigators decided to continue the study after the 5-year follow-up. They did not find any differences in kidney function or incidence of fractures; however, recurrence of hypercalcemia was only observed in the cinacalcet group, indicating that surgical treatment is more effective over longer periods of time [41].

Another study performed to compare parathyroidectomy and cinacalcet therapy was described by Soliman et al. It is an observational, comparative, open-label study analyzing 45 patients taking cinacalcet and 14 patients after parathyroidectomy, over one year. Researchers, contrary to previous studies, did not find any significant changes in laboratory parameters between the two groups; however, patients undergoing surgical treatment required higher doses of vitamin D and calcium supplementation during the first 8 weeks after surgery. This observation indicates that pharmacological management allows for more gradual and reversible control of hyperparathyroidism compared with surgical intervention [45].

There are several retrospective studies comparing those two interventions. The trial performed by Rivelli et al. analyzed 76 patients (46 in the cinacalcet group and 30 in the parathyroidectomy group) and took one year of follow-up. This study demonstrated that cinacalcet effectively controlled serum calcium and phosphate concentrations but was less effective than parathyroidectomy in normalizing PTH levels. On the other hand, this analysis showed significantly better kidney function in patients taking cinacalcet [46]. Finnerty et al. conducted a study in which they observed more frequent normalization of PTH levels and fewer renal allograft failures in participants who underwent parathyroidectomy. This is the reason why the authors recommend considering surgical treatment in patients having an inadequate level of PTH 1 year after kidney transplantation [47]. PTH level, measured 4 weeks after transplantation, can also serve as a biomarker for the occurrence of hypercalcemic hyperparathyroidism; monitoring its level should be a crucial part of complex therapy after kidney transplantation [48]. Another retrospective analysis was performed by Dulfer et al. and included 30 patients who underwent parathyroidectomy and 64 patients taking cinacalcet, from two academic centers. Investigators noticed that serum calcium normalized in both groups; however, only patients from parathyroidectomy group also had normalized levels of PTH—taking into consideration that elevated PTH level after kidney transplantation increases the risk of allograft dysfunction. This study shows that the surgical approach is a favorable way of treatment to avoid long-term consequences. There were no differences between the two groups in renal function, but three patients from the parathyroidectomy group had persistent hyperparathyroidism. For this reason, they had to undergo reoperation; however, it is impossible to compare with AEs of patients from the cinacalcet group, as it was not reported in this study [49]. The last study in this section, conducted by Jung et al. in a single center, included 64 patients after parathyroidectomy and 19 patients taking cinacalcet. Surgical treatment resulted in lower serum calcium and significantly lower PTH level; however, there were also higher levels of creatinine and higher rates of graft rejection. That is why it is necessary to closely observe renal parameters in patients after parathyroidectomy [50]. Comparison of effect of cinacalcet and parathyroidectomy in kidney transplantation patients is shown in Table 2.

Surgical treatment frequently demonstrates greater biochemical normalization in comparative studies, but evidence may be influenced by selection bias, as surgical cohorts are often younger with fewer comorbidities. Cinacalcet is, in some studies, associated with better renal function; however, evidence remains limited. There is a need to perform a large, multi-center prospective RCT to make a coherent comparison of both methods, as the superiority of surgical treatment is not universal and depends on patient profile. Effect of cinacalcet use in RCTs are presented in Table 3.

In kidney transplant patients, parathyroidectomy is more effective than cinacalcet in normalizing PTH and calcium levels and improving bone mineralization in the long term, although it is associated with a higher risk of hypocalcemia, hungry bone syndrome, and postoperative complications [43,45]. Cinacalcet is considered a safe alternative providing calcium and phosphorus control with better preservation of transplanted kidney function in certain cohorts. Despite adverse events being relatively common and current evidence remaining heterogeneous, it should be considered as a treatment option for patients who are not candidates for parathyroidectomy [44].

## 6. Conclusions

Direct comparison of cinacalcet and parathyroidectomy is difficult, as available data are heterogeneous, with variable study duration, patient profile, and co-administered drugs. Cinacalcet appears effective option in lowering serum level of PTH, calcium, and phosphorus in patients suffering from all stages of CKD, but long-term data about patients’ outcomes is still evolving. Use of this drug in combination with vitamin D analogs and phosphorus binders can improve biochemical parameters without increasing the risk of side effects. The most common AEs are gastrointestinal problems such as nausea or vomiting, and hypocalcemia; however, they are generally transient and mild. In the case of hypocalcemia, it is easily manageable by use of vitamin D analogs. Cinacalcet may be preferred in clinically stable patients tolerating oral therapy, especially when minimizing the risk of hypocalcemia is important. In contrast, etelcalcetide can be recommended in patients with poor adherence to oral drug administration and gastrointestinal symptoms; however, the higher incidence of hypocalcemia requires monitoring of calcium levels. In patients receiving multiple cytochrome P450-metabolized drugs, etelcalcetide offers an advantage due to its lack of CYP-mediated interactions. Despite all described benefits, meta-analyses did not prove improvement in survival of patients. Etelcalcetide may reduce non-adherence-related treatment failure, but cinacalcet can be preferred in systems prioritizing outpatient flexibility and lower drug acquisition cost. Currently, there are no unified guidelines for treating patients with SHPT and PHPT, but our critical review provides information that treatment should include pharmacotherapy first and surgery for refractory or severe cases. Pharmacological therapy is not a curative method of treatment, in contrast to parathyroidectomy, offering more definitive biochemical normalization, especially in patients suffering from tertiary hyperparathyroidism with severe or persistent hypercalcemia, elevated PTH, or progressive bone disease. Recommendations for the use of cinacalcet in selected groups of patients with CKD are shown in Table 4. Summarizing, every patient needs an individual approach, because heterogeneity among studies makes data synthesis difficult. Further research should involve long-term cardiovascular endpoints, measuring quality of life, and differences in economic effectiveness.

## Figures and Tables

**Table 1 biomedicines-14-00016-t001:** Effect of cinacalcet in hemodialysis patients.

Study	Type	Comparator	Included Trials	Main Outcomes	Bone Parameters
Wang et al. [20]	Meta-analysis	Cinacalcet vs. placebo	19 RCTs	Reduction in serum calcium, phosphorus and PTH	Uncertain effect
Zu et al. [21]	Meta-analysis (RCTs + obs.)	Cinacalcet vs. no cinacalcet	10 RCTs 4 obs.	Reduction in serum calcium, phosphorus and PTH	Increase in BMD in femur bone
Xu et al. [11]	Meta-analysis	Cinacalcet + Vit D vs. Vit D alone	8 RCTs	Reduction in serum calcium and phosphorus	-
Palmer et al. [22]	Network meta-analysis	Cinacalcet vs. other calcimimetic agents	36 trials	Strong reduction in PTH	Uncertain effect
Lozano-Ortega et al. [23]	Bayesian meta-analysis	Cinacalcet vs. Vit D + phosphorus binders	10 RCTs 6 obs.	Reduction in mortality	Lower risk of fractures
Liu Y et al. [24]	Meta-analysis	Cinacalcet vs. placebo	27 RCTs	Reduction in osteocalcin, increase in bone alkaline phosphatase	Increase in mineralization
Liu et X al. [25]	Network meta-analysis	Cinacalcet vs. paricalcitol	21 RCTs	Reduction of calcium–phosphorus product and FGF23	Increase in BMD
Chen et al. [26]	Meta-analysis	Cinacalcet vs. parathyroidectomy	13 cohort studies	Strong reduction in mortality	Increase in BMD
Song et al. [27]	Meta-analysis	Cinacalcet vs. parathyroidectomy	23 RCTs and cohort studies	Reduction in iPTH level and mortality	Increase in BMD

**Table 2 biomedicines-14-00016-t002:** Comparison of cinacalcet and parathyroidectomy in kidney transplant recipients.

Study	Design	Comparator	iPTH Change—Cinacalcet	iPTH Change—PTx	Renal Function
Cruzado et al. [44]	RCT, *n* = 30, 12 months follow-up	Subtotal parathyroidectomy	Mild reduction	Moderate reduction	Reduction in both groups
Moreno et al. [41]	5-year follow-up of Cruzado study	Subtotal parathyroidectomy	No significant change	No significant change	No change
Soliman et al. [43]	Observational, *n* = 59, 12 months follow-up	Subtotal parathyroidectomy	No significant change	No significant change	No change
Rivelli et al. [44]	Retrospective, *n* = 76, 12 months follow-up	Subtotal parathyroidectomy	Moderate reduction	Strong reduction	Increase in cinacalcet group
Finnerty et al. [47]	Retrospective, *n* = 133, 7-year follow-up	Parathyroidectomy vs. cinacalcet + Vit D	Mild reduction	Moderate reduction	Increase in allograft survival (PTx)
Dulfer et al. [49]	Retrospective, *n* = 94, 12 months follow-up	Subtotal parathyroidectomy	Mild reduction	Moderate reduction	No change
Jung et al. [50]	Retrospective, *n* = 83, 12 months follow-up	Subtotal parathyroidectomy	Mild reduction	Moderate reduction	Reduction in PTx group

**Table 3 biomedicines-14-00016-t003:** Comparison of outcomes of cinacalcet use in randomized clinical trials.

Study	Design	Intervention	Results	Adverse Events	Risk of Bias
Cruzado JM et al. [44]	Multicenter, open-label randomized trial in transplant recipients; *n* = 30, 12-month follow-up	Subtotal parathyroidectomy vs. cinacalcet	Greater iPTH reduction in parathyroidectomy group, normocalcemia achieved in all patients from surgical group and 10/15 from cinacalcet group	Post-operative hypocalcemia in surgical treatment group, and gastrointestinal adverse effects in cinacalcet group	Moderate risk: small sample size and open-label
Moreno P et al. [41]	5-year follow-up of the Cruzado study	Subtotal parathyroidectomy vs. cinacalcet	Recurrence of hypercalcemia only in cinacalcet group	Recurrent hypercalcemia in cinacalcet group	High risk: observational, small cohort, no re-randomization
Wang AY et al. [31]	Randomized trial in peritoneal dialysis patients; *n* = 65, 12-month follow-up	Total parathyroidectomy vs. cinacalcet	No significant differences in cardiovascular parameters; improvement of biochemical markers in both groups	Fewer hypercalcemia hospitalizations in cinacalcet group	Moderate risk: randomized design, but limited sample size
Block GA et al. [29]	Randomized, multicenter clinical trial in hemodialysis patients; *n* = 683, 6-month follow-up	IV etelcalcetide vs. oral cinacalcet	Etelcalcetide showed superior PTH lowering	Hypocalcemia more frequent in etelcalcetide group	Low risk: large, multicenter, randomized trial with appropriate methods for drug comparison

**Table 4 biomedicines-14-00016-t004:** Recommendations for the use of cinacalcet in selected groups of patients with chronic kidney disease [51].

CKD Grade Group	Diet	Calcium Supplementation	Cholecalciferol and Vitamin D Analogs	Cinacalcet
G3a-G5 ND	phosphorus-restricted	in case of hypocalcaemia	cholecalciferol recommended, analogs not routinely indicated	not routinely indicated
G5D HD	phosphorus-restricted	in case of hypocalcaemia	analogs recommended	recommended
G5D PD	phosphorus-restricted	in case of hypocalcaemia	analogs recommended	recommended
G5T	phosphorus-controlled	not routinely indicated	cholecalciferol recommended, analogs not routinely indicated	not routinely indicated

## Data Availability

No new data were created or analyzed in this study.

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
