# Peer review of "Cinacalcet Efficacy in Hyperparathyroidism—Chronic Kidney Disease—Non-Dialysis, Hemodialysis, Peritoneal Dialysis, Kidney Transplantation: Critical Review"

_biomedicines, 2025, doi:10.3390/biomedicines14010016_

Round 1
Reviewer 1 Report
Comments and Suggestions for Authors
The authors present a concise mini-review addressing the efficacy of cinacalcet in the management
of secondary and tertiary hyperparathyroidism across the spectrum of chronic kidney disease,
including non-dialysis patients, hemodialysis patients, peritoneal dialysis, and kidney transplant
recipients.
Although the topic has already been explored in previous literature, both older and more recent, it
remains clinically relevant, particularly given the heterogeneity of available studies and the absence
of fully consistent conclusions.
The manuscript is generally well-structured, but several aspects would benefit from clarification or
further elaboration.
Major comments:
1-The manuscript aims to summarize the efficacy of cinacalcet in CKD and hemodialysis patients.
However, the section on hemodialysis lacks a critical comparison with other therapeutic options,
particularly etelcalcetide, which represents the main intravenous alternative during dialysis.
Including such a comparison—regarding efficacy, adverse events, and cost—would considerably
strengthen the review and offer a more complete perspective for clinicians
2- The conclusion section would benefit from a more analytical synthesis rather than a descriptive
summary. In particular, incorporating a brief comparison between cinacalcet and other available
treatments (most importantly etelcalcetide) would provide the reader with a clearer take-home
message.
3- While concise, the introduction omits relevant elements that would contextualize the topic more
effectively.
For example, adding references that could explain the higher risk of hypercalcemia in non-dialysis
patients—especially in relation to vitamin D analog use or self-medication—would improve
completeness. Cases such as the following could be cited for illustration: Pini S, Scaparrotta G, Di
Vico V, Fragasso A, Stefanelli LF, Nalesso F, Calò LA. Vitamin D intoxication induced severe
hypercalcemia from self-medication for COVID-19 infection: a public health problem? Minerva
Endocrinol (Torino). 2022 Sep;47(3):371-374.Additional case reports addressing vitamin D–related
hypercalcemia could also be mentioned.
Minor comments:
Line 50: add an “and” for a better comprehension of the period.
Line 177: I think that is “but” and not “nut”
Line 343: the line need of remodulation of the punctuation and I would add “for this reason” , to
better explain the concept.
Line 358: I think is “with” and not “while”
In general, some acronyms appear without prior definition (e.g., sPTH, tPTH). Introducing them
upon first use would improve clarity. In several sentences the punctuation could be refined; for
example, “for this reason,” could facilitate comprehension in one instance.
Author Response
Dear Reviewer,
Thank You very much for the effort put into the review and for valuable comments. The comments helped us significantly improve the article. Yours Sincerely Authors

Reviewer 2 Report
Comments and Suggestions for Authors
This review by Lewandowski et al examines the evidence for using cinacalcet in different CKD populations. The article is thorough and well-organizes, and presents results from essentially every study published on cinacalcet use. The divisions into non-dialysis, HD, PD, and transplant patients with a summary of studies in each group was quite helpful.
There are minor grammatical mistakes, mostly with the frequent omission of definite (the) and indefinite (a/an) articles, and I would recommend improving this prior to publication. But the presented science is excellent and otherwise I have no recommendations for improvement.
Author Response

(The authors gave the same response as above.)

Reviewer 3 Report
Comments and Suggestions for Authors
Minor revision
This manuscript provides a comprehensive overview of cinacalcet use across different CKD populations. The topic is clinically relevant, and the structure is generally clear. However, several issues need to be addressed before the review can be accepted:
- The manuscript summarizes many studies but does not adequately evaluate heterogeneity, study limitations, or the risk of bias—particularly in observational studies and surgical cohorts.
- Overemphasis on biochemical outcomes. Hard clinical endpoints such as mortality, cardiovascular events, fractures, vascular calcification, and graft function require deeper discussion, especially when RCTs and observational studies provide conflicting results.
- Conclusions such as “surgical treatment is generally superior” or “cinacalcet preserves kidney function better” should be more cautious and acknowledge limitations of the available evidence, selection bias, and surgical risks (e.g., hypocalcemia, hungry bone syndrome).
- Consider adding summary tables comparing outcomes across CKD stages and between cinacalcet vs parathyroidectomy to improve readability.
- Several statements such as “Cinacalcet is safe and well tolerated” “Surgical treatment is generally superior” “Cinacalcet preserves kidney function” should be softened to reflect the limited and heterogeneous evidence.
Author Response

(The authors gave the same response as above.)

Round 2
Reviewer 1 Report
Comments and Suggestions for Authors
I reviewed the revised version (v2) of the submitted mini-review on the efficacy of cinacalcet in the management of secondary and tertiary hyperparathyroidism across the spectrum of chronic kidney disease (non-dialysis CKD, hemodialysis, peritoneal dialysis, and kidney transplant recipients).
Overall, some topic remains clinically relevant. The revised manuscript shows improvements, particularly in the hemodialysis section and in the contextualization of hypercalcemia risk in non-dialysis CKD. However, several issues remain, and in my opinion the manuscript still requires minor revision before it can be considered for publication.
Minor Comments
1) Comparison with alternative therapies in hemodialysis (etelcalcetide)
Status: Addressed (substantially).
The revised version includes a structured comparison between cinacalcet and etelcalcetide, covering key aspects such as biochemical efficacy (PTH reduction thresholds), adverse events (notably hypocalcemia), differences in administration/pharmacokinetics, drug–drug interactions (CYP-related), adherence, and a brief reference to cost-effectiveness measures (e.g., QALY/ICER). This addition strengthens the manuscript and increases its clinical usefulness.
Remaining issue: the referencing format appears inconsistent and should be corrected.
2) Conclusions remain too descriptive and insufficiently analytical
Status: Partially addressed.
* To strengthen the Conclusions, I recommend explicitly stating the main clinical implications—for example, outline when cinacalcet should be preferred over etelcalcetide or vice versa, considering administration route, adherence, tolerability, hypocalcemia risk, potential drug interactions, and cost or organizational factors.
* how these options compare with parathyroidectomy in selected cases (especially in tertiary hyperparathyroidism).
3) Introduction: better contextualization of hypercalcemia risk in non-dialysis CKD
Status: Addressed.
The revised Introduction better contextualizes the risk of hypercalcemia in non-dialysis CKD and appropriately links this risk to vitamin D analogues and self-medication. The inclusion of relevant references (including the suggested case report on vitamin D intoxication during COVID-19) improves the clinical framing of the topic.
Minor Comments
A) Language and typos
Status: Not fully addressed.
Multiple language issues and typographical errors persist and should be carefully corrected throughout the manuscript. Examples include:
* “bnut” instead of “but”;
* awkward phrasing in the Introduction (e.g., formulations such as “and v. Vitamin D analogues …”);
* additional scattered typos and editorial artifacts that reduce readability.
A thorough language revision by a fluent English speaker (or professional editing service) is strongly recommended.
B) Punctuation and style
Status: Not fully addressed.
Several sentences would benefit from revised punctuation to improve logical flow and clarity. Where appropriate, adding short connectors (e.g., “for this reason,” “therefore,” “however”) would help guide the reader through the argument.
C) Acronyms
Status: Partially addressed.
Some acronyms are insufficiently defined at first use (or inconsistently used throughout the text, e.g., SHPT/THPT). Please ensure that all acronyms are defined upon first appearance and used consistently thereafter.
D) Referencing and formatting
Status: Not addressed satisfactorily.
The citation numbering/formatting appears inconsistent (e.g., atypical bracketed reference patterns). Please check and correct:
* reference numbering consistency;
* formatting according to the journal style;
* completeness and accuracy of the bibliography.
Author Response
Dear Reviewer,
The authors thank you very much for the effort put into the review and for valuable comments. The comments helped us significantly improve the article. Yours Sincerely Authors
